# Research on the Deviation of Corporate Green Behaviour under Economic Policy Uncertainty Based on the Perspective of Green Technology Innovation in Chinese Listed Companies

**Deshuai Hou, Luhan Shi** **, Hong He \* and Jian Xiong**

School of Accounting, Capital University of Economics and Business, No. 121 Zhangjia Road, Huaxiang, Fengtai District, Beijing 100070, China; houdeshuai@126.com (D.H.); 13071053170@163.com (L.S.); hsiungchien790421@163.com (J.X.)
\* Correspondence: Heh@cueb.edu.cn; Tel.: +86-136-8151-1878

**Abstract:** In the current context of economic transformation and a complex environment, increasing economic policy uncertainty may lead to deviations in corporate green behaviour, and it is particularly important to correct such deviations. On this basis, this paper empirically analyses the impact of economic policy uncertainty on corporate green behaviour bias based on statistical data of Chinese listed companies from 2007 to 2019. We find that economic policy uncertainty inhibits corporate green technology innovation but increases corporate innovation as a whole. Using the mechanism test, it was found that the internal inducement is mainly due to the prominent financing problems and limited development ability under the influence of uncertainty. After carrying out a heterogeneity test, it was found that economic policy uncertainty causes enterprises to deviate from green technology innovation more significantly in state-owned enterprises and protected industries, while this effect is significantly reduced when firms face fierce product market competition. Furthermore, strengthening executives' power and implementing incentive mechanisms can more effectively correct the deviation. This study provides empirical evidence with which to strengthen corporate green innovation practices.

**Keywords:** economic policy uncertainty; financing constraints; development capacity; green technology innovation

## 1. Introduction

With the rapid development of China's economy, people's living standards have undergone dramatic changes. However, behind this rapid economic development lies the over-exploitation of resources and the destruction of the ecological environment, and this haphazard economic development has become a significant obstacle to the quality and sustainable development of China's economy. In order to solve the contradiction between sustainable development and the requirements for improving people's quality of life, the concept of green development has gradually begun to gain popularity. Since the 18th National Congress of the Communist Party of China, Chinese governments have also attached great importance to the relationship between the protection of the ecological environment and economic development, emphasizing that ecological civilization is a millennium plan for sustainable development. The report of the 19th National Congress of the Communist Party of China put forward the goal of "accelerating the reform of the ecological civilization system and building a beautiful China", which clearly required "building a market-oriented green technology innovation system". The National Development and Reform Commission and the Ministry of Science and Technology jointly issued the Guidance on Building a Market-oriented Green Technology Innovation System in 2019, which defined green technology as an emerging technology with which to reduce consumption and pollution, improve ecology, promote the construction of an ecological civilization, and

achieve harmony between humans and nature. At the 75th session of the United Nations General Assembly, the Chinese government proposed that it would strive to achieve carbon peaking by 2030 and carbon neutrality by 2060. The realization of these carbon emissions targets depends on the upgrading of China's industrial structure and the transformation of its energy structure, but in a final analysis, it depends on green, high-quality development with green behaviour at its core. It can be seen that green, high-quality development has become an inevitable choice for China's sustainable economic development, and green technological innovation is the core driver for green, high-quality development. All social organisations should continue to increase their efforts with respect to green technology innovation and improve the ability of enterprises to innovate independently in order to achieve the transformation of green products from "Made in China" to "Created in China". This will improve the core competitiveness of enterprises while reducing environmental pollution and promote sustainable economic development.

On the one hand, green behaviour relies on market incentives [1] and the needs of enterprises' own development strategies [2]; on the other hand, it may be closely related to the external policy environment. At present, the economic policies of major economies around the world have been adjusted to some degree, and China is no exception, but this also aggravates economic policy uncertainty to a certain extent. In particular, in recent years, the international economic situation has become increasingly complex; frequent trade conflicts and the COVID-19 pandemic have forced economic policies to remain in a state of instability. Economic policy uncertainty is an important factor in the uncertainty of the external environment that affects firms [3], which may have a profound impact on enterprise innovation, especially green innovation. Some studies have found that economic policy uncertainty can motivate firms to improve innovation, strengthen their internal development [4], and increase enterprises' R&D innovation through selection effects and incentive effects [5]. However, other studies have found that economic policy uncertainty can lead to a shift from manufacturing to service in business, which can lead to a bias in firms' breakthrough innovation behaviour, especially for non-state enterprises [6]. Furthermore, in a global context, the impact of economic policy uncertainty on enterprise innovation may also have an inverted U-shaped feature, which is more significant in developing countries [7]. Therefore, there is no consensus on the impact of economic policy uncertainty on firm innovation in the existing studies; thus, this paper provides a further focus on "altruistic" green innovation behaviour with high "positive externalities" to explore the influence of economic policy uncertainty on enterprise innovation from a new perspective.

Accordingly, this paper uses a traditional OLS model and empirically analyses the impact and action paths of economic policy uncertainty on firms' green innovation behaviour based on statistical data of Chinese listed firms from 2007 to 2019. In addition, based on considerations from the perspectives of a firm's nature, industry characteristics, and a competitive environment, we conduct a heterogeneity analysis of the main effects and propose possible solutions. Compared with previous studies, this paper makes potential theoretical contributions in the following three areas: Firstly, this paper expands the research on green technology innovation factors; existing studies have mainly studied firms' green technology innovation from perspectives such as green credit [1], corporate environmental protection actions [2], environmental regulation tools [8], and financial subsidies [9]. However, given the contradiction between the profit-seeking nature of firms and the positive externality of green technology innovation, firms' green technology innovation may also be affected by the external economic policy environment to a greater extent. Therefore, this paper provides new empirical evidence for the research of enterprises' green technology innovation from the perspective of macroeconomic policy uncertainty. Secondly, although existing studies have found that macroeconomic policy uncertainty stimulates enterprise innovation behaviour to a certain extent [4,5,10,11], which is of great significance with respect to understanding the driving mechanism of enterprise innovation, they mainly focus on the "self-interested" innovation of enterprises, while ignoring "altru-

istic" innovation. Given the heterogeneity of the effects of "self-interested" and "altruistic" innovation on firms' value growth, economic policy uncertainty may also have a bias with respect to firm innovation behaviour. Based on the existing research, this paper further focuses on altruistic green technology innovation with high positive externality, which is of great theoretical value with regard to promoting corporate innovation research. Finally, green technology innovation is the key to implementing high-quality green development. Promoting enterprises' green technology innovation not only requires the consideration of micro-enterprises but also the creation of a stable macroeconomic policy environment. This research conclusion has important practical significance in terms of deepening China's green development and achieving the "double carbon" goal.

The rest of this paper is structured as follows. Section 2 reviews the relevant literature on economic policy uncertainty and green technology innovation and proposes the research hypotheses of this paper. Section 3 introduces the study design of this paper, including data selection, regression models, and variable definitions. Section 4 presents regression results and robustness tests. Section 5 analyses the differences in the relationship between economic policy uncertainty and green technology innovation under different conditions. Finally, Section 6 concludes the paper.

## 2. Literature Review and Research Hypothesis

### 2.1. Literature Review

#### 2.1.1. Influencing Factors of Green Technology Innovation

Braun and Wield (1994) [12] first proposed to incorporate pollution emissions into the concept of green technology on the basis of traditional technology, which attracted widespread scholarly attention. In recent years, with the increasing prominence of ecological and environmental problems in China, the Fifth Plenary Session of the 18th CPC Central Committee put forward the five major development concepts of "innovation, coordination, green, openness and sharing" in 2015, emphasizing that innovation should lead efforts toward green development. The General Secretary has also repeatedly emphasized the green development concept of "green water and green mountains are golden mountains and silver mountains". Green development is the core of national development, and humanity should adhere to sustainable development and form a new, harmonious relationship with nature. As people pay more and more attention to the ecological environment, green technology innovation is becoming an increasingly hot topic for scholars. The literature on the factors influencing green technological innovation mainly concerns the following areas: The first area is environmental policy at the macro level. Appropriate environmental regulation can indirectly enhance enterprise green technology innovation [13]. Command-based regulation and investment-based regulation can promote green technology innovation, while cost-based regulation does not significantly contribute to green technology innovation [14]. Li (2021) [15] also confirmed the inverted U-shaped relationship between environmental regulation and green technology innovation. The second area is media attention at the meso level. Media attention can significantly improve the green technology innovation performance of heavily polluting firms [16], and it only confers a significant positive influence on green technology innovation inputs and promotes green technology innovation outputs when the level of marketization is incorporated [17]. The final area is enterprise behaviour at the micro level. Venture capital, R&D investment [18], and board governance [19] can promote green technology innovation, while environmental protection strategies have a U-shaped relationship with the external benefits of green technology innovation [2].

#### 2.1.2. Economic Consequences of Economic Policy Uncertainty

Economic policy uncertainty refers to the uncertainty generated by the government when formulating or implementing economic policies [20,21]. After the financial crisis, global economic policies have been experiencing instability [22]. The literature on the economic consequences of economic policy uncertainty focuses on two aspects: firstly, at the macro level, economic policy uncertainty not only inhibits economic growth [23] but

also reduces financial stability; however, the degree gradually increases over time in the short term and decreases in the medium and long terms [24]. At the micro level, economic policy uncertainty increases stock price volatility [22] and corporate commercial credit financing [25]. Economic policy uncertainty increases the willingness and size of large shareholders' equity pledges through financing constraints and mispricing [26]. However, Liu et al. (2021) [27] believe that an increase in economic policy uncertainty leads to a significant decline in controlling shareholders' equity pledges. In addition, economic policy uncertainty not only reduces investment opportunities [28] but also enhances firms' investment convergence behaviour [29].

In summary, the literature on the economic consequences of economic policy uncertainty has focused on economic growth, financing behaviour, and corporate share prices and explored this uncertainty's impact on corporate innovation, but there is still no consensus on its impact on green technology innovation. There is a large body of literature on the factors influencing green technology innovation, which mainly focuses on environmental regulation, media attention, and micro-firm behaviour. While economic policy uncertainty is an important source of external environmental information for the high-quality development of enterprises, there is little research on the impact of economic policy uncertainty on the green technology innovation of enterprises. Therefore, this paper explores the impact and influencing mechanisms of economic policy uncertainty with regard to green technology innovation.

### 2.2. Research Hypothesis

Green technological innovation was defined as new processes or products reducing environmental pollution [30,31]. Green technological innovation is becoming the leading force in the new era of technological competition [32]. Green technology innovation can generate positive spillover effects in the stages of innovation and diffusion. The spillover effect in the diffusion stage refers to the lower external cost of green technology innovation compared with other competitive products in the market, while the spillover effect in the innovation stage means that the value created by green technology innovation can be shared by society. Enterprises engaging in green technology innovation cannot receive an income that matches costs, which reduces their enthusiasm toward green technology innovation [33]. Green technology innovation is not only affected by its own characteristics but also by macroeconomic policies. In recent years, the economic situation in China and abroad has been severe, and economic policy uncertainty may be both an opportunity for and a challenge to green technology innovation. Therefore, this paper explores the impact of economic policy uncertainty on green technology innovation with respect to the following aspects.

The business decisions of enterprises are vulnerable to the influence of macroeconomic policies, and volatile economic policies may also hamper companies' ability to cope with complex market changes, forcing them to reduce their green technology innovation behaviour. Firstly, a higher level of economic policy uncertainty can affect management's judgement of an economic situation. When the degree of economic policy uncertainty increases, risk-averse management may reduce willingness and negatively affect behaviour regarding green technology innovation in order to reduce the systemic risk brought on by the market. Intrinsically, the technological innovation of enterprises faces great risks; thus, management is more likely to adopt a risk-averse approach to inhibit enterprise innovation under the dual pressure of higher policy uncertainty [34]. Secondly, shareholder investment is an important source with which firms can engage in green technology innovation. Increased economic policy uncertainty affects investors' judgments about the future development of firms, thus affecting their direct investment in firms. Increased economic policy uncertainty can significantly increase equity risk [35], which indirectly reduces firms' willingness to engage in green technology innovation. Thirdly, the impact of economic policy uncertainty on corporate green technology innovation is also bound to be reflected in funding. Economic policy uncertainty may intensify the information asymmetry between

enterprises and banks or other creditors, thereby increasing the financing difficulty experienced by enterprises and thus reducing innovation investment [36]. Enterprises require a large amount of investment capital for green technology innovation and invest the capital in intangible assets that are difficult to recover within a short time frame [37], which can also reduce management's willingness to invest in green technology innovation. Furthermore, regarding green technology innovation under conditions of economic uncertainty, firms not only face the same risks faced during ordinary innovation but may also be at an even greater disadvantage. For firms affected by economic policy uncertainty, the optimal test is to implement market-incentivised innovations that strengthen a firm's resilience to risk, that is, to render them more likely to implement "survival" innovation rather than "altruistic" green technology innovation with high "positive externality". Therefore, even though green technology innovation has high positive externality with regard to the entirety of society, it is not the optimal strategy for enterprises, at least in the short term and when under the impact of economic policy uncertainty. As a result, under the conditions of economic policy uncertainty, enterprises' degree of general innovation increases significantly [4,5], while enterprise green technology innovation declines significantly, and the game decision result of "individual optimal choice is not group" appears.

In summary, increased economic policy uncertainty makes it more difficult for firms to raise finances and for investors to evaluate green technology innovation projects and offers a more conservative pool of options from which management can choose green technology innovation, which, in turn, reduces willingness and the scale of investment in green technology innovation. Especially under the influence of economic policy uncertainty, enterprises are more likely to implement market incentivised innovation rather than green technology innovation. Therefore, this paper contends that economic policy uncertainty can inhibit firms' green technology innovation. Therefore, this paper proposes research hypothesis H1:

**H1:** *If other conditions remain the same, economic policy uncertainty can inhibit enterprises' green technology innovation.*

Financing constraints can restrict a firm's business-related decision-making process. During periods of economic volatility, enterprises experience difficulty attaining the funds needed for technological innovation activities and may be forced to abandon good investment opportunities [38]. Economic policy uncertainty constrains the development of enterprises in the external environment and raises their financial risk [39]. Moreover, enterprises with stronger financing constraints are more sensitive to a changing economic environment [40]. Therefore, economic policy uncertainty can increase the financing difficulties faced by enterprises. Green technology innovation activities require a great deal of financial support, while enterprises with higher financing constraints find it more difficult to obtain capital, so such enterprises are more inclined to reduce their levels of green technology innovation. Economic policy uncertainty itself has a restraining effect on the green technology innovation of enterprises, while financing constraints provide additional resistance, thereby reducing the green technology innovation of enterprises, which becomes the intermediary bridge that allows economic policy uncertainty to further inhibit firms' green technology innovation. Therefore, this paper contends that economic policy uncertainty inhibits enterprises' green technology innovation by increasing financing constraints. Therefore, this paper proposes research hypothesis H2:

**H2:** *Economic policy uncertainty increases corporate financing constraints and thus inhibits corporate green technology innovation.*

Corporate sustainability refers to the ability of a company to achieve its business objectives in the pursuit of long-term survival and sustainable development, while enabling it to continue to make profits and grow steadily over a significant period of time in an already prominent competitive field and in an expanding business environment in the future. While green technology innovation is an investment activity involving a

relatively long period, it inevitably requires enterprises to be sustainable; otherwise, it is difficult for green technology innovation to succeed. As an external factor, economic policy uncertainty plays an integral role in the development of a company, not only directly affecting its business decisions but also influencing investors' judgement of the company. Firstly, as mentioned in the above analysis, economic policy uncertainty can make it more difficult for companies to raise capital, and a more constrained financing environment is not conducive to improving the sustainability of companies. Secondly, uncertain economic policies can lead to insufficient investment [41], and, in turn, inefficient investment can reduce enterprise value [42], which can also affect a firm's sustainability. Finally, the separation of the two rights makes the interests of shareholders and management inconsistent, while increased economic policy uncertainty further aggravates the degree of information asymmetry between shareholders and management, aggravates the agency problem, and increases agency costs, thus reducing enterprise sustainability. Sustainability is also a key factor influencing the success of green technology innovation; therefore, this paper contends that economic policy uncertainty inhibits green technology innovation by reducing firms' sustainability. Thus, this paper proposes research hypothesis H3:

**H3:** *Economic policy uncertainty reduces corporate sustainability and thus inhibits corporate green technology innovation.*

Main business development ability is an important guarantee with respect to enterprises' ability to carry out green technology innovation, which is a risky investment activity. Only when main business development is stable can technological innovation run smoothly. On the one hand, economic policy uncertainty encourages enterprises to take greater risks [43]. Executives with a risk preference may invest in financial assets with higher risks in order to obtain high returns, while the financialization of an enterprise can damage its main business development ability. On the other hand, economic policy uncertainty can increase financial systemic risk [44] and stock price crash risk [45], while stock price fluctuation also affects enterprises' main business development ability. Therefore, when economic policy uncertainty increases, enterprises need to take more market risks than they would in a stable economic period, while economic policy uncertainty threatens enterprises' main business development ability, thus affecting enterprises' green technology innovation. In conclusion, economic policy uncertainty can deteriorate the main business development ability of an enterprise, which is an important factor for the success of green technology innovation. Thus, this paper contends that main business development ability plays a mediating role in the inhibitory effect of economic policy uncertainty on green technology innovation. Therefore, this paper proposes research hypothesis H4:

**H4:** *Economic policy uncertainty hampers corporate main business development ability and thus inhibits corporate green technology innovation.*

### 3. Research Design

*3.1. Sample Selection and Data Sources*

This paper uses panel data of Chinese listed companies from 2007 to 2019 to study the impact of economic policy uncertainty on corporate green technology innovation. The micro data were obtained from the CSMAR and Wind databases, and the macro data were obtained from the official website of the National Bureau of Statistics. We filtered the data according to the following principles: (1) remove ST enterprises with two consecutive years of operating losses and special treatment and remove *ST enterprises with three consecutive years of operating losses and delisting warnings; (2) exclude financial and insurance enterprises; and (3) eliminate enterprises with abnormal or missing financial data. We winsorized the main continuous variables at 1% and 99% levels to avoid extreme values affecting the results. Finally, we obtained 25,300 annual observations of 3197 enterprises.

*3.2. Empirical Model*

Regarding Hypothesis 1, this paper establishes an econometric model to test the impact of economic policy uncertainty on green technology innovation, as shown in Formula (1):

$$LnGrepatent_{it} = \beta_0 + \beta_1 EPU_t + \beta Control_{it} + year_t + industry_i + \varepsilon_{it} \tag{1}$$

In model (1), $LnGrepatent_{it}$ is an explained variable, which represents green technology innovation; $EPU_t$ is an explanatory variable, which represents economic policy uncertainty; $Control_{it}$ is the control variable of this paper; $year_t$ represents time-fixed effects; $industry_i$ represents industry-fixed effects; $\varepsilon_{it}$ represents the error term; and the subscripts $i$ and $t$ represent enterprise and year, respectively. The regression results of this paper were estimated using the Ordinary Least Squares (OLS) method, which is one of the most fundamental forms of regression analysis for parameter estimation in linear regressions. OLS regression requires the least amount of model conditions and has the advantages of computational simplicity, practical flexibility, and wide applicability, so this method is used to explore the impact of economic policy uncertainty on green technology innovation. Additionally, this paper tests its research hypotheses using panel data and adjusts standard errors for potential heteroskedasticity and within-group serial correlation bias by employing clustering at the firm level. We account for the signs and significance of $\beta_1$ in model (1). If $\beta_1$ is significantly negative, it indicates that economic policy uncertainty inhibits enterprises' green technology innovation, which verifies research hypothesis H1 of this paper.

*3.3. Description of Variables*

3.3.1. Explained Variable

Green technology innovation (*LnGrepatent*). Drawing on the relevant research of Tao et al. (2021) [46], we mainly use the sum of the number of applications for green invention patents and green utility model patents to measure green technology innovation; the number of green invention patents and the number of green utility model patent applications are based on the CNRDS database. In addition, the sum of the green invention patent and green utility model patent applications is first added to 1 and then taken as the natural logarithm. The larger the degree of green technology innovation (*LnGrepatent*), the greater the level of green technology innovation a company has achieved.

3.3.2. Explanatory Variable

Economic policy uncertainty (*EPU*). This paper adopts the uncertainty index of China's economic policy developed by Baker et al. (2016) [22]. This index was extracted from public data, such as newspapers, news media, and expert forecast reports, using the text analysis method and was indexed with 100 in January 1995 so as to obtain the monthly data of China's economic policy uncertainty. In this paper, the geometric average of one year is divided by 100 to measure economic policy uncertainty (*EPU*). The higher the *EPU* value, the higher the degree of economic policy uncertainty.

3.3.3. Control Variables

According to previous studies [47–49], company size (*Size*), leverage ratio (*Lev*), proportion of tangible assets (*Tang*), revenue growth rate (*RevGrowth*), monetary capital (*Cash*), board size (*Board*), dual position of chairman and general manager (*Dual*), executive shareholding rate (*Mgh*), proportion of independent directors (*Ddrate*), equity concentration (*Top1*), per capita GDP by province (*lnGRP*), enterprise innovation (*LnTotalpatent*), year, and industry may affect corporate investment and further impact green technology innovation. Thus, this study controls for these variables. The variables' definitions are shown in Table 1.

**Table 1.** Variable definitions.

| Variable Types | The Variable Name | Variable Symbol | Variable Definitions |
|---|---|---|---|
| Explained variable | Green technology innovation | *LnGrepatent* | The sum of green invention patent and green utility model patent applications is first added to 1 and then taken as the natural logarithm |
| Explanatory variable | Economic policy uncertainty | *EPU* | A monthly index of China's economic policy uncertainty/12/100 |
| Control variables | Company size | *Size* | The natural log of total assets |
| | Leverage ratio | *Lev* | Total liabilities/total assets |
| | Proportion of tangible assets | *Tang* | (total assets—intangible assets)/total assets |
| | Revenue growth rate | *RevGrowth* | (current year's operating income—last year's operating income)/last year's operating income |
| | Monetary capital | *Cash* | Monetary capital/total assets |
| | Board size | *Board* | The natural log of the number of board members |
| | Dual position of chairman and general manager | *Dual* | If the chairman and general manager are the same person, this value is equal to 1; otherwise, it is equal to 0 |
| | Executive shareholding rate | *Mgh* | Number of shares held by senior executives/total number of shares at the end of the year |
| | Proportion of independent directors | *Ddrate* | Number of independent directors/number of board members |
| | Equity concentration | *Top1* | Number of shares held by the largest shareholder/total number of shares at the end of the year |
| | Per capita GDP by province | *lnGRP* | The natural log of the per capita GDP of the province where the enterprise is located |
| | Enterprise innovation | *LnTotalpatent* | The natural log of the total number of corporate patent applications is first added to 1 and then taken as the natural logarithm |
| | Year | *year* | Control year factor |
| | Industry | *Industry* | Control industry factor |

## 4. Empirical Results and Analysis

### 4.1. Descriptive Statistics of Variables

Table 2 provides descriptive statistics of the main variables used in this paper. It can be seen from Table 2 that, firstly, the mean value of green technology innovation (*LnGrepatent*) is 0.270, the standard deviation is 0.676, the minimum value is 0.000, and the maximum value is 3.401, indicating that there are great differences in the green innovation behaviours of various enterprises in the sample period. Secondly, the mean value of the economic policy uncertainty Index (*EPU*) is 2.939, the standard deviation is 2.114, the minimum value is 0.947, and the maximum value is 7.793, indicating that there is a peak of policy adjustment in the sample period and that economic policy uncertainty changes greatly. In other words, the main variables of this paper are significantly different in the sample period, which indicates that this study has certain research significance and practical value. In addition, the control variables are largely within the normal range and will not be detailed herein. In this paper, a multicollinearity test of the main variables showed that there was no multicollinearity problem among the variables and thus the specific results are not listed beyond this point.

**Table 2.** Descriptive statistics of main variables.

| Variable | Observation | Mean | SD | Median | Min | Max |
|---|---|---|---|---|---|---|
| LnGrepatent | 25,300 | 0.270 | 0.676 | 0.000 | 0.000 | 3.401 |
| EPU | 25,300 | 2.939 | 2.114 | 2.380 | 0.947 | 7.793 |
| Size | 25,300 | 22.120 | 1.294 | 21.940 | 19.660 | 26.100 |
| Lev | 25,300 | 0.437 | 0.210 | 0.431 | 0.053 | 0.935 |
| Tang | 25,300 | 0.953 | 0.052 | 0.966 | 0.673 | 1.000 |
| RevGrowth | 25,300 | 0.193 | 0.461 | 0.114 | −0.557 | 3.138 |
| Cash | 25,300 | 0.180 | 0.128 | 0.145 | 0.014 | 0.634 |
| Board | 25,300 | 2.143 | 0.199 | 2.197 | 1.609 | 2.708 |
| Dual | 25,300 | 0.250 | 0.433 | 0.000 | 0.000 | 1.000 |
| Mgh | 25,300 | 0.091 | 0.164 | 0.000 | 0.000 | 0.648 |
| Ddrate | 25,300 | 0.373 | 0.053 | 0.333 | 0.333 | 0.571 |
| Top1 | 25,300 | 0.349 | 0.152 | 0.333 | 0.044 | 0.750 |
| lnGRP | 25,300 | 10.990 | 0.533 | 11.040 | 9.660 | 12.010 |
| LnTotalpatent | 25,300 | 2.319 | 1.762 | 2.398 | 0.000 | 6.739 |

*4.2. Regression Results*

Table 3 lists the regression results regarding the effect of economic policy uncertainty on green technology innovation. Column (1), column (2), and column (3) list the regression results without control variables, for the control year and industry only, and for the sum of all the control variables, respectively. The results show that the regression coefficients of economic policy uncertainty (*EPU*) are significantly negative at the 1% confidence level, which verifies research hypothesis H1, indicating that economic policy uncertainty can inhibit green technology innovation and lead to deviation in firms' green innovation behaviour.

**Table 3.** Baseline regression results regarding the effect of economic policy uncertainty on green technology innovation.

| Variable | LnGrepatent | | |
|---|---|---|---|
| | **(1)** | **(2)** | **(3)** |
| EPU | −0.016 *** | −0.015 *** | −0.035 *** |
| | (−8.63) | (−6.40) | (−6.72) |
| Size | | | 0.041 *** |
| | | | (3.88) |
| Lev | | | 0.094 ** |
| | | | (2.21) |
| Tang | | | 0.100 |
| | | | (0.81) |
| RevGrowth | | | −0.037 *** |
| | | | (−5.05) |
| Cash | | | 0.206 *** |
| | | | (3.34) |
| Board | | | 0.071 |
| | | | (1.23) |

**Table 3.** *Cont.*

| Variable | LnGrepatent | | |
|---|---|---|---|
| | **(1)** | **(2)** | **(3)** |
| *Dual* | | | 0.012 |
| | | | (0.73) |
| *Mgh* | | | 0.038 |
| | | | (0.83) |
| *Ddrate* | | | 0.170 |
| | | | (1.01) |
| *Top1* | | | −0.062 |
| | | | (−1.03) |
| *lnGRP* | | | 0.001 |
| | | | (0.07) |
| *LnTotalpatent* | | | 0.150 *** |
| | | | (20.34) |
| *year* | No | Yes | Yes |
| *Industry* | No | Yes | Yes |
| *_cons* | 0.317 *** | −0.003 | −1.277 *** |
| | (21.51) | (−0.08) | (−3.40) |
| *Observations* | 25,300 | 25,300 | 25,300 |
| *Adjusted R*$^2$ | 0.002 | 0.115 | 0.251 |

Note: Values in parentheses are t values after adjusting for standard error. **, and *** refer to statistical significance at 10%, and 5%, respectively.

Based on the analysis of hypothesis H2, the financing constraints are used as the intermediary variable in this paper and, drawing on the method of Hadlock and Pierce (2010) [50], we use the *SA* index to measure enterprises' financing constraints (*FC*). The calculation formula for the *SA* index is as follows:

$$SA = -0.737 \times Size + 0.043\,Size^2 - 0.04 \times Age \tag{2}$$

where *Size* is a firm's size; *Age* is the natural logarithm of the observation year of the enterprise minus the establishment year of the enterprise sample plus 1; and *SA* index represents the financing constraint (*FC*). The following model was developed to test the mediation effect, and $Medium_{it}$ represents the mediation variables. It has been verified that economic policy uncertainty can inhibit green technology innovation. If $\beta_1$ of model (3) is significantly positive, this means that economic policy uncertainty can increase enterprise financing constraints, and if $\beta_2$ of model (4) is significantly negative, this indicates that financing constraints play a mediating role in the relationship between economic policy uncertainty and green technology innovation.

$$Medium_{it} = \beta_0 + \beta_1 EPU_t + \beta Control_{it} + year_t + industry_i + \varepsilon_{it} \tag{3}$$

$$LnGrepatent_{it} = \beta_0 + \beta_1 EPU_t + \beta_2 Medium_{it} + \beta Control_{it} + year_t + industry_i + \varepsilon_{it} \tag{4}$$

The regression results are shown in columns (2) and (3) of Table 4. In column (2), the regression coefficient of economic policy uncertainty (*EPU*) is significantly positive at a 1% confidence level, indicating that economic policy uncertainty increases firms' financing constraints. In column (3), the regression coefficient of economic policy uncertainty (*EPU*) is significantly negative at a 1% confidence level, and the regression coefficient of financing constraints (*FC*) is significantly negative at a 1% confidence level, indicating that financing

constraints play a partial mediating role in the relationship between economic policy uncertainty and green technology innovation. The Sobel test is a commonly used test for determining the significance of a causal effect. This method assumes that the effect of an independent variable on a dependent variable can be decomposed into a direct effect and an indirect effect, for which the Sobel statistic is used to measure the significance of the indirect effect. Therefore, we refer to the study of Sobel (1982) [51] and use the Sobel test to test the significance of the mediating effect in this paper. The results of the Sobel test are shown in Table 4. The Z statistic of the Sobel test is −12.39, which is significantly negative at the 1% confidence level; thus, the mediating effect of the financing constraint passes the test. Therefore, the deviation of enterprises' green innovation behaviour caused by economic policy uncertainty is mainly restricted by financing constraints, which verifies research hypothesis H2.

**Table 4.** The intermediary effect of financing constraints and enterprises' sustainability.

| Variable | *LnGrepatent* | *FC* | *LnGrepatent* | *SustainGrowth* | *LnGrepatent* |
|---|---|---|---|---|---|
| | (1) | (2) | (3) | (4) | (5) |
| EPU | −0.035 *** | 0.012 *** | −0.031 *** | −0.002 *** | −0.035 *** |
| | (−6.72) | (9.09) | (−6.74) | (−3.03) | (−6.65) |
| FC | | | −0.353 *** | | |
| | | | (−2.83) | | |
| SustainGrowth | | | | | 0.091 * |
| | | | | | (1.78) |
| Size | 0.041 *** | −0.012 *** | 0.037 *** | 0.016 *** | 0.040 *** |
| | (3.88) | (−3.42) | (3.87) | (13.45) | (3.71) |
| Lev | 0.094 ** | −0.025 ** | 0.085 ** | −0.102 *** | 0.103 ** |
| | (2.21) | (−2.14) | (1.97) | (−13.32) | (2.33) |
| Tang | 0.100 | 0.042 | 0.115 | 0.067 *** | 0.094 |
| | (0.81) | (1.16) | (0.93) | (3.37) | (0.76) |
| RevGrowth | −0.037 *** | 0.009 *** | −0.034 *** | 0.052 *** | −0.042 *** |
| | (−5.05) | (4.58) | (−4.76) | (24.48) | (−5.71) |
| Cash | 0.206 *** | −0.075 *** | 0.181 *** | 0.054 *** | 0.201 *** |
| | (3.34) | (−5.30) | (2.89) | (6.85) | (3.26) |
| Board | 0.071 | −0.008 | 0.067 | −0.001 | 0.071 |
| | (1.23) | (−0.76) | (1.18) | (−0.09) | (1.23) |
| Dual | 0.012 | −0.010 *** | 0.009 | −0.000 | 0.012 |
| | (0.73) | (−3.23) | (0.54) | (−0.03) | (0.73) |
| Mgh | 0.038 | −0.051 *** | 0.020 | 0.020 *** | 0.036 |
| | (0.83) | (−5.81) | (0.43) | (4.01) | (0.79) |
| Ddrate | 0.170 | −0.204 *** | 0.097 | −0.048 ** | 0.175 |
| | (1.01) | (−5.27) | (0.58) | (−2.50) | (1.04) |
| Top1 | −0.062 | −0.029 ** | −0.071 | 0.042 *** | −0.065 |
| | (−1.03) | (−2.36) | (−1.21) | (6.51) | (−1.08) |
| lnGRP | 0.001 | −0.013 *** | −0.003 | −0.000 | 0.001 |
| | (0.07) | (−2.64) | (−0.14) | (−0.02) | (0.06) |

**Table 4.** *Cont.*

| Variable | LnGrepatent | FC | LnGrepatent | SustainGrowth | LnGrepatent |
|---|---|---|---|---|---|
| | (1) | (2) | (3) | (4) | (5) |
| LnTotalpatent | 0.150 *** | −0.004 *** | 0.149 *** | 0.005 *** | 0.150 *** |
| | (20.34) | (−2.88) | (20.49) | (6.66) | (20.32) |
| year | Yes | Yes | Yes | Yes | Yes |
| Industry | Yes | Yes | Yes | Yes | Yes |
| _cons | −1.277 *** | 3.641 *** | 0.007 | −0.349 *** | −1.245 *** |
| | (−3.40) | (33.43) | (0.02) | (−8.72) | (−3.29) |
| Observations | 25,300 | 25,298 | 25,298 | 25,298 | 25,298 |
| Adjusted R² | 0.251 | 0.102 | 0.254 | 0.144 | 0.251 |
| Sobel test Z value | | −12.39 | | | −10.06 |

Note: Values in parentheses are t values after adjusting standard error. *, **, and *** refer to statistical significance at 10%, 5%, and 1%, respectively.

Based on the analysis of hypothesis H3, we use sustainability as a mediating variable to test for the mediating effect and use the sustainable growth rate to measure sustainability (*SustainGrowth*), which was obtained from the CSMAR database, and we apply model (4) and model (5) to determine the mediating effect. If $\beta_1$ of model (4) is significantly negative, this indicates that economic policy uncertainty reduces sustainability, and if $\beta_2$ of model (5) is significantly positive, this indicates that economic policy uncertainty inhibits firms' green technology innovation by reducing sustainability. The regression results are shown in columns (4) and (5) of Table 4. In column (4), the regression coefficient of economic policy uncertainty (*EPU*) is significantly negative at the 1% confidence level, indicating that economic policy uncertainty reduces sustainability. In column (5), the regression coefficient of economic policy uncertainty (*EPU*) is significantly negative at the 1% confidence level, and the regression coefficient of sustainability (*SustainGrowth*) is significantly positive at the 10% confidence level, suggesting that sustainability plays a partial mediating role in the relationship between economic policy uncertainty and green technology innovation. In addition, the results of the Sobel test are shown in Table 4. The Z statistic of the Sobel test is −10.06, which is significantly negative at the 1% confidence level; thus, the mediating effect of sustainability passes the test. Therefore, economic policy uncertainty leads to deviations in green innovation behaviour mainly by reducing firms' sustainability, which verifies research hypothesis H3.

Based on the analysis of hypothesis H4, we referred to the research of Hu et al. (2015) [52] and Du et al. (2017) [53] and used the difference between the return on total assets and the return on investment regarding financial assets to measure main business development ability (*CorePerf*). The specific calculation formulae for the two indicators are as follows: ① CorePerf1 = (operating profit-investment income-return on fair value changes + return on investment with respect to associates and joint ventures)/total assets; ② CorePerf2 = (total profit-investment income-return on fair value changes + return on investment with respect to associates and joint ventures)/total assets. This paper uses model (4) and model (5) to test the mediation effects. If $\beta_1$ of model (4) is significantly negative, this shows that economic policy uncertainty reduces the main business development ability of the enterprise, and if $\beta_2$ of model (5) is significantly positive, this indicates that economic policy uncertainty inhibits enterprises' green technology innovation capacity by reducing the main business development ability of the enterprise.

The regression results are shown in Table 5. The regression coefficients of economic policy uncertainty (*EPU*) in columns (2) and (4) are significantly negative at the 1% confidence level, indicating that economic policy uncertainty reduces the main business development ability of the enterprise. The regression coefficient of CorePerf1 in column (3) is signifi-

cantly positive at the 10% confidence level, and the regression coefficient of CorePerf2 in column (5) is significantly positive at the 5% confidence level. These results show that main business development ability plays a partial mediating role in the relationship between economic policy uncertainty and green technology innovation. In addition, the results of the Sobel test are shown in Table 5. The z-statistics for the two Sobel tests are −19.11 and −39.03, which are significantly negative at the 1% confidence level; thus, the mediating effect of main business development ability passes the tests. Therefore, economic policy uncertainty leads to deviations in green innovation behaviour by reducing enterprises' main business development ability, which verifies research hypothesis H4.

**Table 5.** The intermediary effect of enterprise main business development ability.

| Variable | LnGrepatent | CorePerf1 | LnGrepatent | CorePerf2 | LnGrepatent |
|---|---|---|---|---|---|
| | **(1)** | **(2)** | **(3)** | **(4)** | **(5)** |
| EPU | −0.035 *** | $-7.37 \times 10^7$ *** | −0.041 *** | $-9.11 \times 10^7$ *** | −0.040 *** |
| | (−6.72) | (−4.27) | (−6.51) | (−5.06) | (−6.48) |
| CorePerf1 | | | 0.000 * | | |
| | | | (1.95) | | |
| CorePerf2 | | | | | 0.000 ** |
| | | | | | (2.21) |
| Size | 0.041 *** | $6.87 \times 10^8$ *** | 0.032 ** | $7.38 \times 10^8$ *** | 0.029 ** |
| | (3.88) | (14.26) | (2.51) | (14.62) | (2.25) |
| Lev | 0.094 ** | $-1.27 \times 10^9$ *** | 0.072 | $-1.22 \times 10^9$ *** | 0.074 |
| | (2.21) | (−8.75) | (1.49) | (−8.00) | (1.54) |
| Tang | 0.100 | $-7.17 \times 10^8$ | 0.123 | $-7.16 \times 10^8$ | 0.125 |
| | (0.81) | (−1.57) | (0.86) | (−1.51) | (0.88) |
| RevGrowth | −0.037 *** | $8.98 \times 10^7$ *** | −0.035 *** | $8.21 \times 10^7$ *** | −0.035 *** |
| | (−5.05) | (4.72) | (−3.87) | (4.17) | (−3.88) |
| Cash | 0.206 *** | $4.80 \times 10^8$ *** | 0.215 *** | $5.30 \times 10^8$ *** | 0.213 *** |
| | (3.34) | (2.72) | (2.91) | (2.85) | (2.88) |
| Board | 0.071 | $1.30 \times 10^7$ | 0.063 | $1.09 \times 10^7$ | 0.063 |
| | (1.23) | (0.09) | (0.96) | (0.07) | (0.96) |
| Dual | 0.012 | $9.00 \times 10^7$ ** | 0.007 | $8.90 \times 10^7$ ** | 0.007 |
| | (0.73) | (2.12) | (0.34) | (2.04) | (0.33) |
| Mgh | 0.038 | $1.08 \times 10^8$ | −0.008 | $1.08 \times 10^8$ | −0.008 |
| | (0.83) | (1.36) | (−0.14) | (1.32) | (−0.14) |
| Ddrate | 0.170 | $1.41 \times 10^9$ *** | 0.026 | $1.51 \times 10^9$ *** | 0.019 |
| | (1.01) | (2.61) | (0.14) | (2.72) | (0.10) |
| Top1 | −0.062 | $3.84 \times 10^8$ ** | −0.132 * | $3.63 \times 10^8$ ** | −0.132 * |
| | (−1.03) | (2.20) | (−1.91) | (2.00) | (−1.92) |
| lnGRP | 0.001 | $1.24 \times 10^8$ ** | −0.016 | $1.49 \times 10^8$ ** | −0.017 |
| | (0.07) | (1.98) | (−0.64) | (2.28) | (−0.69) |
| LnTotalpatent | 0.150 *** | $8.09 \times 10^7$ *** | 0.153 *** | $8.66 \times 10^7$ *** | 0.152 *** |
| | (20.34) | (4.65) | (17.87) | (4.81) | (17.86) |
| year | Yes | Yes | Yes | Yes | Yes |

**Table 5.** *Cont.*

| Variable | LnGrepatent | CorePerf1 | LnGrepatent | CorePerf2 | LnGrepatent |
|---|---|---|---|---|---|
| | (1) | (2) | (3) | (4) | (5) |
| Industry | Yes | Yes | Yes | Yes | Yes |
| _cons | −1.277 *** | −1.53 × 10$^{10}$ *** | −0.823 ** | −1.67 × 10$^{10}$ *** | −0.735 * |
| | (−3.40) | (−11.76) | (−2.11) | (−12.21) | (−1.91) |
| Observations | 25,300 | 16,554 | 16,554 | 16,554 | 16,554 |
| Adjusted R$^2$ | 0.251 | 0.370 | 0.272 | 0.384 | 0.273 |
| Sobel test Z value | | −19.11 | | −39.03 | |

Note: Values in parentheses are t values after adjusting standard error. *, **, and *** refer to statistical significance at 10%, 5%, and 1%, respectively.

*4.3. Robustness Test*

In order to enhance the robustness of the research conclusions, the following robustness tests were conducted.

4.3.1. Instrumental Variable Method

China's economic policy uncertainty has a strong exogenous character [54]. It is difficult for an enterprise's green technology innovation to have a significant impact on macroeconomic policies, so it is less likely that there is a reverse causal relationship between economic policy uncertainty and green technology innovation. In order to enhance the reliability of its conclusions, this paper adopts an instrumental variable method to solve the endogeneity problem. Environmental uncertainty is defined as the rate of change and variability in an organization's external environment [55], the most important elements of which are customers, competitors, government regulations, and unions. Early studies argued that the most appropriate characteristics for measuring a firm's environment were market and technological characteristics (R&D and capital expenditure). However, since then, a greater amount of research has argued that technological characteristics may be more appropriate for larger and more traditional firms (e.g., mining, manufacturing, etc.) as a measure of environmental uncertainty [56,57] and that elements of technological characteristics are also subject to management decisions. To avoid the latter problem, we have only considered the market characteristics of a firm, i.e., the coefficient of variation of sales (CZ), which captures well the environmental uncertainty faced by a firm [58,59]. Dechow (1994) [60] used the coefficient of variation of sales to measure environmental uncertainty. Thus, the coefficient of variation of sales is strongly related to the environmental policy uncertainty faced by firms, where the coefficient of variation of sales measures the degree of dispersion of firms' sales and is not correlated with firms' green innovation behaviour, i.e., this variable satisfies the two prerequisites of the instrumental variables approach. Accordingly, this paper refers to Ghosh and Olsen (2009) [61] and Geng and Guo (2021) [62] and uses the coefficient of variation of sales as an indicator of uncertainty in an environment and as an instrumental variable for economic policy uncertainty, which is calculated as follows:

$$CZ_i = \sqrt{\sum_{k=1}^{n} \frac{(z_i - \bar{z})^2}{n}} / \bar{z} \tag{5}$$

where $CZ_i$ represents the annual-adjusted uncertainty environment index, $z_i$ represents the sales revenue of an enterprise, $z$ represents the average sales revenue of an enterprise, and $k$ represents year. Since the instrumental variables approach is implemented through two OLS regressions, it is referred to as Two-Stage Least Squares (2SLS for short). In brief, the first stage of the 2SLS regressions divides economic policy uncertainty into two components, where one part is the fitted value of economic policy uncertainty and the other part is the component associated with the disturbance term. The regression of green technology

innovation with respect to the fitted value of economic policy uncertainty is then sought in the second stage; that is, the regression of economic policy uncertainty with the endogeneity component is eliminated so that consistent estimates can be obtained. Table 6 shows the test results for the 2SLS method. Column (1) shows that the uncertainty environmental indicator (*CZ*) is significantly correlated with economic policy uncertainty (*EPU*), and column (2) shows that the uncertainty environmental indicator (*CZ*) is not significantly correlated with green technology innovation (*LnGrepatent*). These findings show that the instrumental variable satisfies the two prerequisites of the instrumental variable method. Columns (3) and (4) show that the regression coefficient of economic policy uncertainty (EPU) is still significantly negative in the first stage and second stage of the 2SLS regressions, which is consistent with the research results in Table 3. Therefore, the instrumental variable method nullifies the potential endogenous problem, which further verifies the research conclusions of this paper.

**Table 6.** Endogeneity problem.

| Variable | *EPU* | *LnGrepatent* | | |
|---|---|---|---|---|
| | (1) | (2) | (3) | (4) |
| *CZ* | 0.000 *** | 0.002 | | |
| | (22.43) | (0.14) | | |
| *EPU* | | | −0.032 * | −0.035 *** |
| | | | (−1.65) | (−6.73) |
| *Size* | 0.000 *** | 0.041 *** | 0.040 *** | 0.041 *** |
| | (13.69) | (3.88) | (3.66) | (3.88) |
| *Lev* | −0.000 *** | 0.094 ** | 0.095 ** | 0.094 ** |
| | (−8.19) | (2.22) | (2.20) | (2.22) |
| *Tang* | −0.000 * | 0.103 | 0.104 | 0.103 |
| | (−1.77) | (0.83) | (0.84) | (0.83) |
| *RevGrowth* | −0.000 *** | −0.036 *** | −0.036 *** | −0.036 *** |
| | (−3.08) | (−5.01) | (−4.97) | (−5.02) |
| *Cash* | −0.000 *** | 0.205 *** | 0.207 *** | 0.206 *** |
| | (−6.79) | (3.31) | (3.34) | (3.33) |
| *Board* | −0.000 *** | 0.071 | 0.072 | 0.071 |
| | (−8.90) | (1.24) | (1.24) | (1.24) |
| *Dual* | 0.000 *** | 0.013 | 0.012 | 0.013 |
| | (3.97) | (0.75) | (0.74) | (0.75) |
| *Mgh* | 0.000 *** | 0.038 | 0.037 | 0.038 |
| | (6.49) | (0.82) | (0.78) | (0.83) |
| *Ddrate* | −0.000 | 0.170 | 0.170 | 0.169 |
| | (−1.06) | (1.01) | (1.01) | (1.01) |
| *Top1* | −0.000 *** | −0.062 | −0.060 | −0.062 |
| | (−9.17) | (−1.02) | (−0.96) | (−1.03) |
| *lnGRP* | 0.000 *** | 0.002 | −0.001 | 0.002 |
| | (28.69) | (0.07) | (−0.03) | (0.07) |
| *LnTotalpatent* | 0.000 | 0.150 *** | 0.150 *** | 0.150 *** |
| | (0.29) | (20.35) | (20.24) | (20.38) |

**Table 6.** *Cont.*

| Variable | EPU | LnGrepatent | | |
|---|---|---|---|---|
| | **(1)** | **(2)** | **(3)** | **(4)** |
| *year* | Yes | Yes | Yes | Yes |
| *Industry* | Yes | Yes | Yes | Yes |
| *_cons* | 1.731 *** | −1.342 *** | −1.262 *** | −1.279 *** |
| | $(1.21 \times 10^{12})$ | (−3.51) | (−3.16) | (−3.41) |
| *Observations* | 25,286 | 25,286 | 25,286 | 25,286 |
| *Adjusted $R^2$* | 1.000 | 0.251 | 0.251 | 0.251 |

Note: Values in parentheses are t values after adjusting standard error. *, **, and *** refer to statistical significance at 10%, 5%, and 1%, respectively.

### 4.3.2. Adjusting the Measurement Method of Key Variables

Considering that the measurement results of a single variable may be contingent and contain errors, this paper re-estimates economic policy uncertainty and green technology innovation. Firstly, we used the research conducted by Baker et al. (2016) [22] to construct a monthly index of China's economic policy uncertainty and calculate the arithmetic mean within one year to re-measure the economic policy uncertainty index. As shown in column (1) of Table 7, the regression coefficient of *EPU1* is still significantly negative. Secondly, economic policy uncertainty (*EPU*) was treated with one-stage lag, and a regression was carried out. As shown in Column (2) of Table 7, the regression coefficient of *EPU_l* is still significantly negative. Thirdly, the number of green invention patent applications and the number of green utility model patent applications were used to measure green technology innovation. As shown in Columns (3) and (4) of Table 7, the regression coefficient of *EPU* is still significantly negative. Finally, we used the number of corporate patent applications, excluding green patent applications, to measure corporate innovation. The result is shown in column (5) of Table 7. The regression coefficient of *EPU* is significantly positive at the 1% confidence level, indicating that economic policy uncertainty can promote the implementation of enterprise market-incentivized innovation but also reduce an enterprise's green technology innovation and that the positive externality of corporate innovation is weakening. Therefore, we have further verified the robustness of the conclusions.

**Table 7.** Robustness tests (concerning the adjustment of the measurement method of key variables).

| Variable | LnGrepatent | | lnGIP | lnGUP | LnRepatent |
|---|---|---|---|---|---|
| | **(1)** | **(2)** | **(3)** | **(4)** | **(5)** |
| *EPU* | | | −0.022 *** | −0.027 *** | 0.003 *** |
| | | | (−5.16) | (−7.63) | (3.84) |
| *EPU1* | −0.034 *** | | | | |
| | (−6.72) | | | | |
| *EPU_l* | | −0.066 *** | | | |
| | | (−8.35) | | | |
| *Size* | 0.041 *** | 0.044 *** | 0.039 *** | 0.025 *** | 0.004 *** |
| | (3.88) | (3.93) | (4.48) | (3.56) | (3.25) |
| *Lev* | 0.094 ** | 0.085 * | 0.068 ** | 0.049 * | −0.011 * |
| | (2.21) | (1.86) | (1.98) | (1.73) | (−1.73) |
| *Tang* | 0.100 | 0.062 | 0.014 | 0.172 ** | 0.019 |
| | (0.81) | (0.47) | (0.14) | (2.06) | (0.86) |

**Table 7.** *Cont.*

| Variable | LnGrepatent | | lnGIP | lnGUP | LnRepatent |
|---|---|---|---|---|---|
| | (1) | (2) | (3) | (4) | (5) |
| *RevGrowth* | −0.037 *** | −0.039 *** | −0.026 *** | −0.023 *** | 0.000 |
| | (−5.05) | (−5.18) | (−4.43) | (−4.47) | (0.33) |
| *Cash* | 0.206 *** | 0.247 *** | 0.187 *** | 0.056 | −0.036 *** |
| | (3.34) | (3.47) | (3.60) | (1.41) | (−3.67) |
| *Board* | 0.071 | 0.084 | 0.060 | 0.044 | −0.001 |
| | (1.23) | (1.34) | (1.24) | (1.13) | (−0.22) |
| *Dual* | 0.012 | 0.011 | 0.012 | 0.007 | −0.003 |
| | (0.73) | (0.60) | (0.87) | (0.60) | (−1.03) |
| *Mgh* | 0.038 | 0.033 | −0.002 | 0.055 * | −0.010 |
| | (0.83) | (0.62) | (−0.06) | (1.79) | (−1.22) |
| *Ddrate* | 0.170 | 0.258 | 0.141 | 0.145 | 0.031 |
| | (1.01) | (1.43) | (1.04) | (1.30) | (1.62) |
| *Top1* | −0.062 | −0.065 | −0.068 | 0.005 | 0.008 |
| | (−1.03) | (−1.01) | (−1.38) | (0.12) | (1.01) |
| *lnGRP* | 0.001 | 0.006 | 0.011 | −0.008 | −0.002 |
| | (0.07) | (0.27) | (0.62) | (−0.56) | (−0.81) |
| *LnTotalpatent* | 0.150 *** | 0.152 *** | 0.111 *** | 0.085 *** | 0.994 *** |
| | (20.34) | (19.72) | (18.26) | (16.86) | (1006.69) |
| *year* | Yes | Yes | Yes | Yes | Yes |
| *Industry* | Yes | Yes | Yes | Yes | Yes |
| *_cons* | −1.275 *** | −1.421 *** | −1.197 *** | −0.817 *** | −0.084 ** |
| | (−3.39) | (−3.44) | (−3.75) | (−3.26) | (−2.01) |
| *Observations* | 25,300 | 21,890 | 25,300 | 25,300 | 25,210 |
| *Adjusted* $R^2$ | 0.251 | 0.256 | 0.218 | 0.198 | 0.996 |

Note: Values in parentheses are t values after adjusting standard error. *, **, and *** refer to statistical significance at 10%, 5%, and 1%, respectively.

### 4.3.3. Adjusting the Regression Model

In this paper, we mainly used OLS regression models to empirically test the impact of economic policy uncertainty on enterprises' green technology innovation. Considering the limitations of a single-regression model, we use a fixed effects model to carry out a regression in this section. The results are shown in column (1) of Table 8. The regression coefficient of economic policy uncertainty (*EPU*) is still significantly negative at the 10% confidence level, indicating the reliability of the results. The Tobit model is a type of model in which, although the dependent variable is approximately and continuously distributed as positive values, a proportion of observations that assume a value of zero with a positive probability is contained. The range of values of green technology innovation consists of numbers greater than or equal to 0, and a portion of the enterprises' green technology innovation assumes the value of 0, which meets the requirements of the Tobit model; therefore, we also used the Tobit model for regression. The results are shown in column (2) of Table 8. The regression coefficient of economic policy uncertainty (*EPU*) remains significantly negative at the 1% confidence level, again illustrating the robustness of the results.

**Table 8.** Robustness tests (concerning the adjustment of the regression model).

| Variable | *LnGrepatent* | |
|---|---|---|
| | **(1) Fixed Effects Model** | **(2) Tobit Model** |
| *EPU* | −0.020 * | −0.035 *** |
| | (−1.86) | (−9.27) |
| *Size* | −0.013 | 0.041 *** |
| | (−1.55) | (9.50) |
| *Lev* | 0.067 ** | 0.094 *** |
| | (2.27) | (4.06) |
| *Tang* | 0.118 | 0.100 |
| | (1.06) | (1.32) |
| *RevGrowth* | −0.017 *** | −0.037 *** |
| | (−3.45) | (−4.54) |
| *Cash* | 0.029 | 0.206 *** |
| | (0.78) | (6.20) |
| *Board* | 0.042 | 0.071 *** |
| | (1.00) | (3.01) |
| *Dual* | −0.004 | 0.012 |
| | (−0.32) | (1.37) |
| *Mgh* | −0.081 | 0.038 |
| | (−1.57) | (1.48) |
| *Ddrate* | 0.118 | 0.170 ** |
| | (0.86) | (2.05) |
| *Top1* | −0.112 ** | −0.062 ** |
| | (−2.11) | (−2.39) |
| *lnGRP* | −0.011 | 0.001 |
| | (−0.18) | (0.15) |
| *LnTotalpatent* | 0.109 *** | 0.150 *** |
| | (18.29) | (51.15) |
| *year* | Yes | Yes |
| *Industry* | No | Yes |
| *_cons* | 0.198 | −1.277 *** |
| | (0.31) | (−8.44) |
| *var (e.LnGrepatent)* | — | 0.341 *** |
| *N* | 25,300 | 25,300 |
| *Adjusted $R^2$* | 0.103 | — |

Note: Values in parentheses are t values after adjusting standard error. *, **, and *** refer to statistical significance at 10%, 5%, and 1%, respectively.

## 5. Further Research

### *5.1. Heterogeneity Test*

5.1.1. Nature of Property Rights

The different natures of enterprises can have different impacts on their business behaviour. State-owned enterprises do not blindly pursue economic performance (such enterprises have a greater social responsibility), while private enterprises are more con-

cerned with their own development ability. Therefore, state-owned enterprises and private enterprises may have differences with respect to green technology innovation. The operating decisions of state-owned enterprises are more strongly dependent on economic policies than private enterprises [63], and the government may also have a bias towards state-owned enterprises when formulating economic policies. When economic policy uncertainty is high, firms face higher risks, and state-owned enterprises may be more reluctant to engage in green technology innovation because of the government's sheltering and the enterprises' social responsibility. Therefore, this paper speculates that the effect of economic policy uncertainty on firms' deviation from green technology innovation behaviour is more significant in state-owned enterprises.

In this paper, the sample is divided into state-owned enterprises and private enterprises according to the nature of their respective property rights. If the sample enterprise is a state-owned enterprise, *Soe* is 1; otherwise, it is 0, and we cross economic policy uncertainty (*EPU*) with the nature of property rights (*Soe*). If the regression coefficient of *EPU* × *Soe* is significantly negative, this indicates that the inhibitory effect of economic policy uncertainty on green technology innovation is more significant in state-owned enterprises than in private enterprises. The regression results are shown in Column (1) of Table 9. The regression coefficient of *EPU* × *Soe* is significantly negative at the 5% confidence level, which verifies the speculation made in this section. This is because state-owned enterprises have poor market flexibility and are more likely to be directly affected by economic policies. Therefore, the impact of economic policy uncertainty on enterprises' deviation from green technology innovation behaviour is more obvious for state-owned enterprises.

**Table 9.** Heterogeneity test.

| Variable | *LnGrepatent* | | |
| --- | --- | --- | --- |
| | **(1) Nature of Property Rights** | **(2) Protected Industries** | **(3) Product Market Competition** |
| *EPU* | −0.031 *** | −0.033 *** | −0.041 *** |
| | (−5.40) | (−6.56) | (−7.65) |
| *Soe* | 0.031 | | |
| | (1.16) | | |
| *EPU× Soe* | −0.009 ** | | |
| | (−2.23) | | |
| *Prot* | | 0.098 | |
| | | (1.26) | |
| *EPU × Prot* | | −0.013 ** | |
| | | (−2.14) | |
| *Marcompetition* | | | −0.144 |
| | | | (−1.02) |
| *EPU × Marcompetition* | | | 0.069 *** |
| | | | (4.03) |
| *Size* | 0.041 *** | 0.041 *** | 0.043 *** |
| | (3.77) | (3.87) | (4.00) |
| *Lev* | 0.094 ** | 0.094 ** | 0.098 ** |
| | (2.09) | (2.21) | (2.29) |
| *Tang* | 0.091 | 0.098 | 0.096 |
| | (0.71) | (0.79) | (0.76) |

**Table 9.** *Cont.*

| Variable | LnGrepatent | | |
| --- | --- | --- | --- |
| | **(1) Nature of Property Rights** | **(2) Protected Industries** | **(3) Product Market Competition** |
| *RevGrowth* | −0.036 *** | −0.036 *** | −0.037 *** |
| | (−4.73) | (−5.02) | (−5.12) |
| *Cash* | 0.218 *** | 0.209 *** | 0.206 *** |
| | (3.40) | (3.39) | (3.35) |
| *Board* | 0.074 | 0.070 | 0.068 |
| | (1.22) | (1.22) | (1.17) |
| *Dual* | 0.011 | 0.012 | 0.012 |
| | (0.62) | (0.71) | (0.70) |
| *Mgh* | 0.040 | 0.037 | 0.035 |
| | (0.81) | (0.80) | (0.75) |
| *Ddrate* | 0.180 | 0.169 | 0.161 |
| | (1.02) | (1.00) | (0.96) |
| *Top1* | −0.063 | −0.062 | −0.059 |
| | (−1.02) | (−1.04) | (−0.98) |
| *lnGRP* | −0.002 | 0.001 | 0.003 |
| | (−0.08) | (0.06) | (0.12) |
| *LnTotalpatent* | 0.156 *** | 0.150 *** | 0.151 *** |
| | (20.35) | (20.33) | (20.37) |
| *year* | Yes | Yes | Yes |
| *Industry* | Yes | Yes | Yes |
| *_cons* | −1.284 *** | −1.272 *** | −1.300 *** |
| | (−3.30) | (−3.39) | (−3.42) |
| *Observations* | 23,658 | 25,300 | 25,091 |
| *Adjusted $R^2$* | 0.254 | 0.251 | 0.252 |

Note: Values in parentheses are t values after adjusting standard error. **, and *** refer to statistical significance at 10%, and 5%, respectively.

5.1.2. Protected Industries

Different industry characteristics may also have different effects on business behaviour. Enterprises in government-protected industries are likely to have better conditions for survival, with government support and preferential policies providing better assurance of long-term growth. Under the preconditions of government protection, such enterprises are more likely to maintain their status quo and are less likely to engage in green technology innovation activities; in addition, green technology innovation requires intensive technical expertise, while most enterprises in government-protected industries belong to polluting industries, and the core competitiveness of such industries is not technological innovation. Therefore, this paper speculates that the impact of economic policy uncertainty on firms' deviation from green technology innovation is more significant in protected industries.

This paper draws on relevant studies conducted by Aharony and Wong (2000) [64] and Meng and Shi (2017). According to the securities regulatory commission's "industry classification guidance of listed companies" (2012 edition), the extractive industry (B), petroleum-processing and coking industry (C25), the ferrous-metal-smelting and rolling-processing industry (C31), the non-ferrous-metal-smelting and rolling-processing industry (C32), and the electricity, gas, and water production and supply industry (D) are defined

as protected industries (Prot). If an enterprise belongs to the protected industry, Prot is 1; otherwise, it is 0. We cross economic policy uncertainty (*EPU*) with protected industries (*Prot*). If the regression coefficient of *EPU × Prot* is significantly negative, this indicates that the inhibitory effect of economic policy uncertainty on green technology innovation is more significant in protected industries. The regression results are shown in Column (2) of Table 9. The regression coefficient of *EPU × Prot* is significantly negative at the 5% confidence level, which verifies the speculation made in this section. This is because for protected industries, the government itself provides great protection, and the industries are more likely to maintain the status quo and less likely to carry out green technological innovation. Therefore, the effect of economic policy uncertainty causing firms to deviate from green innovation behaviour is more obvious in protected industries.

### 5.1.3. Product Market Competition

Different product market competition environments can also have different effects on the business behaviour of enterprises. Market structure and degree of competition can affect enterprise risk management decisions [65]. The higher the degree of external product market competition, the greater the competitive pressure enterprises face. To avoid being eliminated, enterprises are more likely to invest in high-risk technological innovation activities to pursue market shares and advantageous market positions. Consequently, in the context of high economic policy uncertainty, enterprises are more motivated to pursue high-risk green technology innovation. Therefore, this paper speculates that fierce product market competition can reduce the impact of economic policy uncertainty, leading to firms' deviation from green innovation behaviour.

Referring to the relevant research of Chen and Ma (2021) [66], this paper incorporates "rate of sales expenses to operating revenue" to measure product market competition (*Marcompetition*). The larger the value of Marcompetition, the more intense the product market competition an enterprise faces. We cross economic policy uncertainty (*EPU*) with product market competition (*Marcompetition*). If the regression coefficient of *EPU × Marcompetition* is significantly positive, this indicates that intense product market competition can weaken the inhibitory effect of economic policy uncertainty on green technology innovation. The regression results are shown in column (3) of Table 9. The regression coefficient of *EPU × Marcompetition* is significantly positive at the 1% confidence level, which verifies the speculation made in this section. The reason behind the latter finding is as follows: when an enterprise is in a fierce product market competition environment, if the enterprise wants to stand out in this market, it must carry out technological innovation, especially the green technological innovation advocated by the state, to obtain a competitive position in the market. Therefore, the impact of economic policy uncertainty causing firms to deviate from green innovation behaviour is significantly reduced when firms are faced with fierce product market competition.

### 5.2. Extensive Test

Corporate governance plays an important role in the business activities of an enterprise. On the one hand, when the chairman and general manager of an enterprise are the same person, the combination of the two positions gives executives more power and can enhance the effectiveness of communication and management decisions. When economic policy uncertainty increases, executives need to make timely judgments to the future development and investment plans of their company. Economic policy uncertainty means that companies are exposed to greater risk, and the combination of the two positions means that executives have more autonomy, which encourages them to choose high-risk green technology innovation activities. In addition, according to agency theory, the separation of the two rights is more likely to lead to principal–agent problems between shareholders and management, and the integration of the two roles encourages senior executives to attach greater importance to the long-term interests of the enterprise, thus promoting the engagement of enterprises in green technology innovation activities. On the other hand, when economic policy uncertainty increases, senior executives would first consider their

own interests based on their precautionary motive, such as increasing the level of corporate cash holdings and reducing high-risk investment in technological innovation in order to reduce the likelihood of dismissal. Executive equity incentives motivate executives to move from being "cautious" to being "risk takers" and to choose risky green technology innovation, even in an environment of high economic policy uncertainty. Therefore, this paper speculates that the combination of two jobs and executive equity incentives can alleviate the inhibitory effect of economic policy uncertainty on green technology innovation and thus play a better role in correcting bias.

In this paper, "whether the chairman and the general manager are the same person" is used to measure the combination of the two positions (Dual). If the chairman and the general manager are the same person, Dual is 1; otherwise, it is 0. Executive equity incentives (*Mgh*) are measured as "the number of shares held by senior executives/the total number of shares at the end of the year". The higher the value of Mgh, the greater the equity incentive for executives. We cross economic policy uncertainty (*EPU*) with the integration of two jobs (*Dual*) and executive equity incentives (*Mgh*), respectively. If the regression coefficients of *EPU* × *Dual* and *EPU* × *Mgh* are both significantly positive, this indicates that the integration of two jobs and executive equity incentives can alleviate the inhibitory effect of economic policy uncertainty on green technology innovation. The regression results are shown in Table 10. The regression coefficient of *EPU* × *Dual* is significantly positive at the 10% confidence level, and the regression coefficient of *EPU* × *Mgh* is significantly positive at the 1% confidence level, thereby verifying the speculation in this section. When the chairman and general manager are the same person, senior executives have greater autonomy in business decisions and are more likely to carry out green technology innovation activities from the perspective of the long-term development of the enterprise. Furthermore, executive equity incentives are more likely to stimulate the enthusiasm of executives and prompt executives to invest in high-risk green technological innovation activities. Therefore, strengthening executive power and implementing equity incentives can positively influence the correction of enterprises' deviation from green innovation behaviour caused by economic policy uncertainty.

**Table 10.** Extensive test.

| Variable | LnGrepatent | |
| :---: | :---: | :---: |
| | **(1)** | **(2)** |
| *EPU* | −0.036 *** | −0.038 *** |
| | (−6.93) | (−7.12) |
| *EPU× Dual* | 0.007 * | |
| | (1.66) | |
| *EPU× Mgh* | | 0.037 *** |
| | | (3.43) |
| *Dual* | −0.008 | 0.012 |
| | (−0.33) | (0.73) |
| *Mgh* | 0.038 | −0.077 |
| | (0.82) | (−1.16) |
| *Size* | 0.041 *** | 0.042 *** |
| | (3.89) | (3.94) |
| *Lev* | 0.093 ** | 0.089 ** |
| | (2.20) | (2.10) |
| *Tang* | 0.099 | 0.096 |
| | (0.80) | (0.77) |

**Table 10.** *Cont.*

| Variable | LnGrepatent | |
| --- | --- | --- |
| | **(1)** | **(2)** |
| *RevGrowth* | −0.037 *** | −0.037 *** |
| | (−5.05) | (−5.05) |
| *Cash* | 0.209 *** | 0.215 *** |
| | (3.37) | (3.46) |
| *Board* | 0.070 | 0.071 |
| | (1.23) | (1.23) |
| *Top1* | −0.062 | −0.064 |
| | (−1.04) | (−1.06) |
| *Ddrate* | 0.169 | 0.171 |
| | (1.01) | (1.02) |
| *lnGRP* | 0.001 | 0.001 |
| | (0.05) | (0.04) |
| *LnTotalpatent* | 0.150 *** | 0.150 *** |
| | (20.33) | (20.35) |
| *year* | Yes | Yes |
| *Industry* | Yes | Yes |
| *_cons* | −1.267 *** | −1.272 *** |
| | (−3.37) | (−3.39) |
| *Observations* | 25,300 | 25,300 |
| *Adjusted $R^2$* | 0.251 | 0.251 |

Note: Values in parentheses are t values after adjusting standard error. *, **, and *** refer to statistical significance at 10%, 5%, and 1%, respectively.

## 6. Conclusions

### 6.1. Research Findings

Recently, green and high-quality development has received increasing attention from scholars, and green technology innovation is a key support for achieving the "double carbon" strategic goal and promoting green and high-quality development. However, although the literature has explored the impact of macroeconomic policy uncertainty on firms' innovation behaviour, it has mainly focused on "self-interested" innovation, while ignoring "altruistic" green technology innovation with high positive externalities. To address this issue, we empirically analysed the impact of economic policy uncertainty on firms' green innovation behaviour based on statistical data on Chinese listed firms from 2007 to 2019. Our findings reveal that there is a significant negative relationship between economic policy uncertainty and green technology innovation, indicating that economic policy uncertainty inhibits corporate green technology innovation. After implementing the instrumental variable method and performing a robustness test, this conclusion was still valid, indicating that economic policy uncertainty is an important factor leading to an enterprise's deviation from green behaviour. In addition, we explored the impact mechanism of economic policy uncertainty on green technology innovation and found that economic policy uncertainty increases firms' financing constraints, reduces their sustainability and main business development ability, and thus leads to deviations from green behaviour. Moreover, based on the perspectives relating to the nature of a firm, industry characteristics, and competitive markets, we conducted a heterogeneity analysis of the main effects and found that economic policy uncertainty leads to deviations from green technology inno-

vation behaviour more significantly in state-owned enterprises and protected industries, and this effect is significantly reduced when enterprises face fierce product market competition. Finally, we found that strengthening executive power and implementing incentive mechanisms can more effectively influence the correction of deviations.

*6.2. Policy Implications*

This paper not only provides a theoretical basis for the development of policies for enhancing green innovation behaviour but also empirical evidence for strengthening green innovation practices in the face of uncertainty. Our research has three practical implications for Chinese companies. Firstly, from the perspective of specific green technology innovation activities, an uncertain economic environment can have a negative effect on an enterprise. The current economic situation in China is characterized by both risks and opportunities, and government departments should build a stable policy system and economic environment to effectively ensure the introduction and implementation of policies and promote their positive effects on enterprises' green behaviour. Secondly, in the process of adjusting economic policies, the government should launch targeted policies to ease enterprises' financing constraints and improve enterprises' sustainability and main business development ability, such as providing corresponding government subsidies and tax incentives for enterprises' specific green technological innovation activities. In addition, regarding the different impacts of economic policy uncertainty on green innovation behaviour in different industries and enterprises, the government should strengthen the guidance of green technology innovation according to the industry and enterprise's own characteristics in order to promote green and high-quality development. Thirdly, Enterprises need to improve the quality of corporate governance in view of the moderating effect of the integration of two jobs and executive equity incentives. On the one hand, enterprises should appropriately increase the power of executives to improve the effectiveness of management decisions regarding green innovation behaviour, which can help increase green technology innovation. On the other hand, enterprises should appropriately strengthen executive equity incentives and ensure that shareholders and management maintain the same goal. Enterprises should consider green technology innovation from a long-term perspective and promote corporate governance mechanisms so that they can more effectively influence the correction of deviations, which can help achieve the goal of China's high-quality economic development.

*6.3. Research Drawbacks and Further Research*

Although this study has important theoretical and practical implications, it still has some drawbacks, which require further research. Firstly, due to the availability of data, we have only considered data from Chinese companies. Therefore, future studies can draw on relevant data from other countries to extend the findings of this paper to such countries. Secondly, regarding the measurement of economic policy uncertainty, we have used the Chinese economic policy uncertainty index developed by Baker et al. (2016) [22], which is a single-measure index whose use might have led to bias in the findings of this paper, so we will continue to explore this measurement method in the future. Thirdly, the green innovation behaviour of enterprises is complex, and not only must the level of green innovation be considered but also green innovation performance and the green innovation effect; thus, determining a way in which to continue to track green innovation behaviour is an issue for further research.

**Author Contributions:** Conceptualization, D.H. and H.H.; Methodology, L.S. and J.X.; Formal analysis, J.X.; Writing—original draft preparation, L.S.; Writing—review & editing, D.H and H.H. All authors have read and agreed to the published version of the manuscript.

**Funding:** This work was supported by the Ministry of Education's Humanities and Social Sciences Research Youth Fund Project "Theoretical and Empirical Research on IPO Registration System Reform and Capital Market Efficiency: Evidence from Regulatory Marketization Reform" (Approval No.: 21YJC630167), and Science and Technology Innovation Project for Doctoral Students of Capital University of Economics and Business (Approval No.: 2023KJCX067).

**Data Availability Statement:** The data presented in this study are available on request from the corresponding author.

**Conflicts of Interest:** The authors declare no conflict of interest.

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
