# Peer review of "Research on the Deviation of Corporate Green Behaviour under Economic Policy Uncertainty Based on the Perspective of Green Technology Innovation in Chinese Listed Companies"

_sustainability, doi:10.3390/su15097611_

Round 1

Reviewer 1 Report

Thank you for the opportunity to review the present study that deals with an interesting and relevant topic of the impact of the economic policy uncertainty on green innovation in China. However, despite the potential relevance of its contributions, I believe that it presents some issues that need to be tackled. In the following, you can find some suggestions and comments that could be useful in improving the contributions of the study.

Control variables (other than year and industry dummies) should be explained in some details and references provided from previous studies that used the same or similar variables.

Are results in Table 3 estimated by the Fixed Effects estimator?

Why is the instrumental variable CZ a valid instrument? This should be explained.

Please provide references for the Sobel test (page 14).

Which hypothesis is tested in Table 6?

Which estimator is used to obtain results in Table 6?

How is sustainability defined (Table 6)?

Which hypothesis is tested in Table 7?

Which estimator is used to obtain results in Table 7?

Which hypothesis is tested in Table 8?

Which estimator is used to obtain results in Table 8?

The quality of English language is good, some minor mistakes, for instance, lines 716 and 720.

Reviewer 2 Report

I consider that the theme of the work is relevant and current for scientific research, organizations and policies and has the potential for publication after a structuring review. So I consider …

- the structure and organization of the work should be improved so that it is clear, easy and comprehensible to read;

- the title is broader than the content (the study is based on Chinese companies and the word “China” does not appear in the title, abstract or keywords);

- the abstract needs to be re-analyzed according to the content of the article, reflecting, according to the “norms”, the objective of the study, the methodology, brief conclusions, contributions, originality, and implications;

- an introduction needs improvement; Following the aspects mentioned for the summary and which should be more extensive at this point; I suggest structuring these aspects in order to make the work clear to the reader, based on the objective of the investigation. Furthermore, I reinforce that the research gap, which justifies its contribution with theoretical foundations, needs to be discussed (line 82 - 120: difficult to understand, at this point introduction); It is necessary to discuss and mention how the research contributes to the literature, in the strands studied, and to the development of knowledge; Do not forget to include recent and updated studies; Line 77: capitalize the author's name; Line 128: speaks of 7 sections and if you look at the numbering, these are not observed (to be corrected);

- in point 2. Review of the literature and research hypotheses: see line 160 and 187 (numbering), line 170 and many others, where spacing is missing between words (review the entire work); As for the hypotheses raised, in my understanding the 2 that present “…may promote/…may inhibit”, are actually only one. However, when rereading the literature presented and the work up to line 550, it seems to me that I should have raised several hypotheses, or a main hypothesis and several secondary ones (also due to the research design presented, the research methods, tests that presented , “analysis of the mechanism of influence” (line 418, which makes me observe the need for more hypotheses) and “Future research” (line 551);

- "1. Research design” (line 282): information from line 284 and 293 that should be in the abstract and introduction; Line 296, presents formula 1 and explanation of the model. However, the meaning reappears from line 312 and with exaggerated repetition of the acronym/phrase (a very frequent situation throughout the work); Decide whether or not to use acronyms in the body text; In the tables it is suggested improvement in their formatting and presentation and consistency in the writing of the sentences (eg always start in capital letters). In the text, when referring to the “Table”, the same consistency; Line 383, clarify the “2SLS method”.

- “Conclusion”: when reading the first sentence of the two paragraphs, I ask: what is the objective(s)? Which hypotheses are raised? I suggest analysis and revision in view of the whole. Do they reflect the content of the article, the relevance to the audience, the impact and implications for academia, profession, practice, and society? To be verified: Main conclusions in view of the objective and methods applied? Gaps filled? Contributions and originality? Implications?  Limitations? Future research paths?

Reviewer 3 Report

In general, the text is constructed correctly and based on appropriate data sources and research methods. However, the limitations of the study should be more clearly described and a discussion section should be clearly distinguished (to what extent the new research results extend, contradict or confirm the existing scientific knowledge with reference to the publications of specific authors). It is necessary to correct the numbering of individual sections.

The language used is correct, but there are minor spelling errors.

Round 2

Reviewer 1 Report

My comments are addressed at a satisfactory level. 

The English language is fine. 

Reviewer 2 Report

Dear authors,

Congratulations on your work and thank you for accepting my constructive suggestions, which made the work clearer and richer.

Good luck